# Facing Anomalies Head-On: Network Traffic Anomaly Detection via Uncertainty-Inspired Inter-Sample Differences

## Abstract

Network traffic anomaly detection is pivotal in cybersecurity, especially as data volume grows and security requirement intensifies. This study addresses critical limitations in existing reconstruction-based methods, which quantify anomalies relying on intra-sample differences and struggle to detect drifted anomalies. In response, we propose a novel approach, the **Un**certainty-Inspired Inter-Sample **Diff**erences method (UnDiff), which leverages model uncertainty to enhance anomaly detection capabilities, particularly in scenarios involving anomaly drift. By employing evidential learning, the UnDiff model gathers evidence to minimize uncertainty in normal network traffic, enhancing its ability to differentiate between normal and anomalous traffic. To overcome the limitations of intra-sample difference quantification in reconstruction-based methods, we propose a novel anomaly score based on inter-sample uncertainty deviation that directly quantifies the anomaly degree. Benefiting from a concise model design and parameterized uncertainty quantification, UnDiff achieves high efficiency. Extensive experiments on three benchmarks demonstrate UnDiff's superior performance in detecting both undrifted and drifted anomalies with minimal computational overhead. This research contributes to the field of network security by introducing a new uncertainty-based modeling paradigm and a novel uncertainty-inspired anomaly score.

## CCS Concepts

• **Security and privacy → Intrusion detection systems**; • **Information systems → Traffic analysis**.

## Keywords

Network Traffic Anomaly Detection; Uncertainty Quantification; Drifted Anomaly Detection; Zero-Positive Learning

## 1 Introduction

Network traffic anomaly detection, a fundamental component of cybersecurity infrastructure [50], plays a pivotal role in identifying malicious activities across various network environments. As data volumes surge exponentially and security requirements are stringent, precisely identifying anomalous network traffic patterns has emerged as a critical imperative. This capability underpins multiple important applications, including enhancing the stability and reliability of network services [12, 29] and fortifying personal privacy protection mechanisms [23, 27].

Current literature on network traffic anomaly detection predominantly employs a reconstruction-based "zero-positive learning" paradigm [5, 19, 26, 50], which only reconstructs normal network traffic distributions during the training phase, typically leveraging architectures such as auto-encoder [51]. Subsequently, during the inference phase, common practice for evaluating anomaly degrees is to utilize a distance-based metric [4, 14, 31, 45, 50, 51], i.e., samples exhibiting significant distance deviation between their pre- and

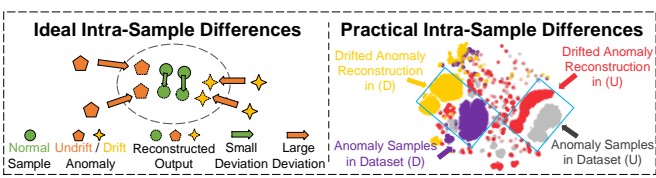

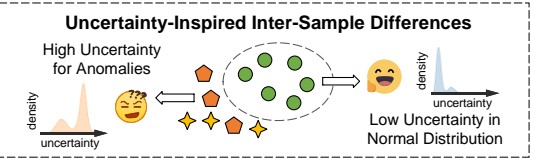

(a) The limitations of existing reconstruction-based methods.

(b) The motivation of our UnDiff.

**Figure 1: Motivation for this work. (a) Existing methods encounter the "identical shortcut" issue, exemplified by the proximity of pre- and post-reconstruction drifted anomalies. (b) Our UnDiff is based on uncertainty-inspired inter-sample differences, facilitating direct anomaly identification.**

post-reconstruction representations are identified as anomalous network traffic, while those demonstrating minimal divergence are considered as normal network traffic (cf. left part of Figure 1(a)).

**Limitations.** Despite the recent advancements in reconstruction-based methods for network traffic anomaly detection, an intrinsic limitation persists. These approaches fully rely on intra-sample differences of pre- and post-reconstruction from an egocentric perspective while insufficiently leveraging inherent inter-sample differences, i.e., the diverse distribution between normal and anomalous traffic [19]. This limitation is exacerbated by the potential "identical shortcut" issue in reconstruction models [42]. Instead of capturing differentiated characteristics of normal and anomalous patterns, reconstruction-based methods tend to converge on a set of shortcut parameters that merely replicate the input as output [30, 32]. This limitation becomes particularly salient in detecting *drifted anomalies*, where the distribution of anomalous data evolves over time. The right part of Figure 1(a) visualizes the pre- and post-reconstruction embedding of drifted anomalies using a state-of-the-art reconstruction-based model Trident [50]. Empirical observations indicate that the pre- and post-reconstruction representations exhibit high proximity in the representation space. The intra-sample differences do not satisfy the ideal institution of reconstruction-based methods, thereby significantly impeding the discrimination of drifted anomalies.

**Contributions.** To address this limitation, we propose a novel **Un**certainty-Inspired Inter-Sample **Diff**erences model (UnDiff). UnDiff leverages the concept of model uncertainty to enhance the discriminative capacity of anomaly detection systems, with particular emphasis on anomaly drift scenarios. As illustrated in Figure 1(b), the key intuition of our model is rooted in the differential uncertainty characteristics exhibited by normal and anomalous traffic patterns. Normal samples, well-represented in the training data,

manifest low model uncertainty. Conversely, anomalous samples, particularly drifted anomalies, induce higher uncertainty due to their deviation from the learned normal patterns [19]. In contrast to existing reconstruction-based methods, our proposed UnDiff addresses the limitation above by directly facilitating inter-sample differences rather than relying on the sub-optimal intra-sample quantification.

Central to our UnDiff is a novel uncertainty learning module that quantifies model detection uncertainty based on informative reconstructed representations. This module employs an evidential learning approach [6], acquiring evidence from training examples to construct an evidential distribution, facilitating robust uncertainty modeling for normal network traffic. Furthermore, we introduce explicit objectives to minimize uncertainty in normal network traffic during training. These objectives provide a more pronounced separation between normal and anomalous samples in the uncertainty space. To overcome the limitation of intra-sample disparity quantification in reconstruction-based methods, we further propose an innovative uncertainty-inspired anomaly score that adequately leverages inter-sample distributional differences for detecting anomalies. Notably, thanks to the concise design of the uncertainty learning module and the efficient parameterized uncertainty quantification technique, the enhancements we proposed above have negligible additional computational overhead. We conduct extensive experiments on three real-world encrypted anomaly traffic datasets and evaluate the performance of UnDiff in both undrifted and drifted anomaly detections. Empirical results verified the effectiveness of our proposed model in detection performance across both scenarios. In summary, our key contributions are threefold:

- We propose a novel uncertainty-based evidential detection framework from an inter-sample difference perspective. Unlike the suboptimal intra-sample difference quantification in existing methods, our approach better utilizes the prior knowledge that anomalies inherently deviate from normal patterns, achieving more effective anomaly detection, particularly in scenarios involving anomaly drift.
- We introduce an innovative uncertainty learning module and a new anomaly score. This module provides an efficient and robust method for capturing sample uncertainty, while the anomaly score effectively quantifies inter-sample differences, significantly enhancing the discriminative capacity of the detection system.
- We conduct comprehensive empirical evaluations on three real-world anomaly network traffic datasets, comparing our approach against several state-of-the-art baselines. The results demonstrate the effectiveness of our framework, UnDiff, in detecting both drifted and undrifted anomalies.

## 2 Related Work

### 2.1 Network Traffic Anomaly Detection

Anomaly detection, particularly zero-positive learning anomaly detection, has gained extensive attention. In this paradigm, only normal data are available during training, and samples that deviate from the learned model behavior are identified as anomalies during inference. Existing methods can be broadly categorized into three groups: distillation-based, statistics-based, and normalizing flow-based approaches [28]. Distillation-based methods focus on intra-sample differences, utilizing a student-teacher architecture to compare the distilled disparities [47, 48]. Conversely, the statistic-based [3, 10] and normalizing flow-based methods [17, 34] aim to learn a mapping from an input domain to a low-dimensional distribution. These approaches quantify inter-sample differences by analyzing deviations in the low-dimensional distribution. However, these methods, primarily designed for natural images, often encounter significant limitations when applied to traffic data. This is due to the unique characteristics of traffic images, such as redundant high-frequency information and disordered texture [51].

**Network Traffic Anomaly Detection.** Current network traffic anomaly detection methods mainly follow a reconstruction-based paradigm. These methods typically reconstruct the normal traffic during training and employ intra-sample differences (i.e., disparities between pre- and post-reconstruction) to identify anomalies. A notable example is GANomaly [4], a prominent reconstruction framework that utilizes a discriminator network to improve normal sample modeling. This approach has been successfully applied to network traffic analysis in subsequent studies [31, 45]. MANomaly [45] introduce a dual autoencoder adversarial training strategy to enhance representation learning, while ARCADE [31] employ WGAN-GP optimization for more effective adversarial training. MFAD [51] identifies a critical "identical shortcut" issue in traffic reconstruction and utilizes low-pass filtering to mitigate this problem. Trident [50] incorporates a U-Net structure to retain more detailed reconstruction information. Most anomaly detection methods for traffic data focus on enhancing the reconstruction quality of normal samples. However, these approaches often evade the "identical shortcuts" issue inherent in reconstruction-based models. To overcome this limitation, we propose a novel paradigm based on inter-sample differences. In contrast to the suboptimal intra-sample differences employed by existing methods, we leverage the prior knowledge that anomalous samples inherently deviate from normal samples, achieving a more effective anomaly identification.

### 2.2 Uncertainty Learning

As deep learning models find increasingly widespread application across diverse domains, accuracy is no longer the only criterion for evaluation. In fields where safety is paramount, there is an urgent need for more trustworthy neural networks. Reliable uncertainty quantification emerges as a critical aspect in this context, as it measures the model's confidence in its output.

As elucidated in the literature [1, 16, 24, 33], two primary categories of uncertainties are associated with neural networks: data uncertainty and model uncertainty. Data uncertainty arises from noise or randomness in the input and can be reduced to zero with sufficient training examples. For model uncertainty, Bayesian learning-based networks provide a mathematically grounded framework, albeit prohibitively expensive to implement and infer. Alternatively, Monte Carlo Dropout [15] approximates Bayesian inference on model parameters. Furthermore, leveraging the ensemble learning paradigm, Deep Ensemble [25] integrates multiple models for uncertainty estimation. To analyze data uncertainty, a unified Bayesian learning-based method [24] has been proposed to directly map

input data to estimations of both data and model uncertainties. Uncertainty learning has also received attention in the field of anomaly detection, with approaches such as Bayesian learning [20] and its variational approximations [18, 21, 22, 43].

Recently, *evidential learning* has emerged as a promising uncertainty quantification approach [20, 36, 38]. This method enables uncertainty estimation in a single model and forward pass with parameterized distributions. In this approach, a neural network outputs the hyperparameters of an evidential distribution, allowing the model to estimate both model and data uncertainties without requiring sampling, thus enhancing the efficiency of uncertainty quantification [6]. However, most existing works on evidential learning are designed for supervised learning in computer vision [20] and necessitate large volumes of labeled data to estimate the uncertainty distribution. This requirement does not fit the typical anomaly detection setting. Therefore, in this study, we explore the application of evidential learning for quantifying the anomaly degree of network traffic in a zero-positive learning context.

## 3 Methodology

In this section, we detail our proposed uncertainty-inspired inter-sample difference method, UnDiff. We describe the research problem and introduce a novel research scenario, anomaly drift. Subsequently, we explicate the requisite data processing modules. We then detail our proposed uncertainty learning module, designed to learn the uncertainty space, thereby facilitating the comparison of inter-sample distribution differences. The schematic representation of our methodological pipeline is illustrated in Figure 2.

### 3.1 Problem Statement

**Network Traffic Anomaly Detection.** This work investigates the zero-positive learning anomaly detection problem in the context of network traffic analysis. Let $\mathcal{X} = \{\mathbf{x}_1, \mathbf{x}_2, ..., \mathbf{x}_N\}$ denote a set of $N$ normal samples, where $\mathbf{x}_i \in \mathbb{R}^d$ is a $d$-dimensional data instance. The objective of detection models is to learn the distributional patterns of normal samples during training. For inference, the model assigns an anomaly score to each test sample $\mathbf{x}_{\text{test}} \in \mathcal{X}_{\text{test}}$, where $\mathcal{X}_{\text{test}}$ represents the set of test samples. This score is derived based on the learned behavior of normal samples. The magnitude of the anomaly score is positively correlated with the likelihood of a sample being identified as anomalous.

**Drifted Anomaly Detection.** The dynamic nature of network activities frequently leads to divergence in the distribution of testing data, a phenomenon known as concept drift [14]. This drift often results in the performance degradation of anomaly detection systems [19, 51]. Existing research on concept drift in anomaly detection primarily focuses on two scenarios: *whole drift*, where both normal and anomalous data experience drift [7, 8, 41, 50] and *normal drift*, where only normal data undergoes drift [19]. However, this study addresses a more realistic scenario: *anomaly drift*, wherein only anomalous data experience drift. This scenario is particularly relevant because, in real-world applications, normal network traffic patterns typically exhibit relative stability, whereas anomalous network traffic patterns often change due to the evolution of attack strategies. Consequently, our research emphasizes the generalization capability of the anomaly detection model when

**Table 1: Comparison of drift scenarios in anomaly detection.**

| Setup | Whole Drift | Normal Drift | Anomaly Drift |
|---|---|---|---|
| Training | Normal (A & B) | Normal (A) | Normal (A) |
| Evaluation | Normal (B) Anomalous (B) | Normal (B) Anomalous (A) | Normal (A) Anomalous (A & B) |

confronted with drifts in the distribution of anomalous data. Table 1 outlines the distinctions among three scenarios.

### 3.2 Data Preprocessing

Network traffic fundamentally manifests as a flow format comprising an ordered sequence of packets. In contrast to statistical features designed based on manual heuristics [31, 45], we directly utilize the original traffic packet information for network traffic modeling. This approach circumvents the introduction of bias associated with manually crafted features. A critical consideration in the data processing of network flows is the appropriate representation method, as it significantly influences the detection accuracy and computational overhead. In this study, we employ a Multi-Channel Traffic Image Construction strategy for traffic flow representation. This approach allows for a more comprehensive and nuanced capture of the multidimensional nature of network traffic.

**Multi-Channel Traffic Image Construction.** While image-based single packet processing has been widely adopted in network traffic anomaly detection [13, 51], the potential of flow-level image construction remains largely unexplored. Drawing inspiration from video anomaly detection methodology [46], which addresses spatio-temporal representation tasks, we propose a novel approach to network flow representation. In our method, we formulate each network flow as a multi-channel image analogous to a video frame sequence. The specific channel order is determined by the packets' chronological arrival, preserving the flow's temporal dimension. This approach offers two significant advantages: (i) Dimensional Efficiency: By extending the representation along the channel dimension rather than width and height, we reduce the generation of subsequent high-dimensional feature maps. This design choice ensures enhanced inference speed. (ii) Informative Representation Preservation: Crucially, this approach adequately preserves informative representations as the contextual relationships between packets (represented as multi-channel images), reflecting the spatio-temporal characteristics of network flows. This temporal and spatial information preservation is critical for capturing the nuanced patterns that may indicate anomalies. For each traffic image, we implement a low-pass filtering process to mitigate noise. This step is necessitated by the unique characteristics of traffic images, which, in contrast to natural images, exhibit a chaotic and textureless state [51]. This phenomenon arises from the abundance of high-frequency components inherent in network traffic data. However, these high-frequency components often manifest as detrimental noise, impeding the model's ability to generalize effectively due to the excessive complexity of the information.

### 3.3 Proposed UnDiff

As shown in Figure 2(a), our UnDiff contains two main components: an evidence extractor to extract evidence and a novel uncertainty

 

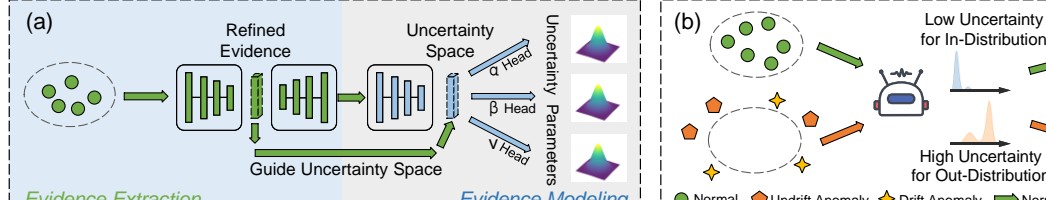

**Figure 2: Overall framework of UnDiff. (a) Training: Uncertainty-Inspired Modeling Process. UnDiff first extracts informative representations from normal network traffic. These refined representations are then utilized to instruct uncertainty parameters for uncertainty representation. (b) Inference: Anomaly Metric Process. UnDiff directly outputs model uncertainty to quantify anomalies by assessing inter-sample differences via a distribution prior that differentiates normal and anomalous traffic.**

learning module to construct an uncertainty space for subsequent anomaly quantification and detection.

### 3.3.1 Evidence Extractor.
The autoencoder is designed to process original input traffic **x**, encoding it into a latent evidence representation **z**, and subsequently decoding it to produce a reconstruction of the input $\hat{\mathbf{x}}$. Given that the input is a multi-channel image, we employ Convolutional Neural Networks (CNNs) as the encoder, following established practices in the literature [51]. It is imperative to note that network traffic inherently comprises a series of packet sequences with distinct spatio-temporal characteristics [49]. However, previous autoencoder-based reconstruction methods have primarily focused on enhancing reconstruction quality, often neglecting the crucial spatio-temporal relationships inherent in the network packets. To address this limitation, we introduce a spatio-temporal aware channel-spatial based attention mechanism, specifically the Convolutional Block Attention Module (CBAM) [40], into our autoencoder architecture. This approach enables us to assign higher importance to significant channel images (temporal features) and spatially relevant regions (spatial features), thereby facilitating the extraction of evidence for critical patterns in multi-channel images. More details of the evidence extractor are in Appendix A.

### 3.3.2 Uncertainty Learning Module.
The predominant zero-positive learning paradigm for anomaly network traffic typically frames this task as a reconstruction problem, optimizing the similarity loss of the original input **x** and the reconstructed output $\hat{\mathbf{x}}$. The intra-sample differences between pre- and post-reconstruction from an egocentric perspective are utilized to quantify the anomaly degrees. However, this paradigm exhibits suboptimal performance due to two primary limitations. Firstly, comparing the differences between samples before and after reconstruction does not directly address the fundamental nature of the problem: anomalous traffic is inherently defined relative to normal traffic patterns. Secondly, the classical "identical shortcut" problem inherent in autoencoder architectures significantly impacts the intra-sample differences of anomalous samples, particularly leading to performance degradation in scenarios involving anomaly drift.

To address these issues concurrently, we propose a novel uncertainty learning module designed to construct an uncertainty space, facilitating direct inter-sample comparisons to detect anomalous network traffic. This module is based on estimating the detection uncertainty, explicitly focusing on model uncertainty, also known

as epistemic uncertainty. Model uncertainty quantifies the uncertainty in estimating model parameters given the training data, effectively measuring the degree of congruity between the model and the data [1]. We posit that this model uncertainty score is intrinsically linked to anomalous patterns and can be leveraged to identify anomalies effectively. The fundamental intuition underpinning our methodology is rooted in the differential uncertainty characteristics exhibited by normal and anomalous traffic patterns [19]. Normal samples, well-represented in the training data, typically manifest low model uncertainty. Conversely, anomalous traffic, particularly in the context of drifted anomalies, induces higher uncertainty due to its deviation from the learned normal patterns.

The uncertainty learning module comprises an encoder and a group of uncertainty parameter heads. The encoder, which shares its architectural design with the preceding encoder of evidence extractor, is based on the reconstruction output $\hat{\mathbf{x}}$. It processes the reconstruction $\hat{\mathbf{x}}$ as input and generates an uncertainty representation $\gamma$, quantifying model's detection uncertainty. The uncertainty parameter heads, implemented as linear layers, translate the uncertainty representation $\gamma$ into their corresponding uncertainty parameters. This transformation facilitates effective uncertainty modeling. Through this mechanism, we explicitly incorporate evidential learning to quantify evidence distribution of normal network traffic. In contrast to Bayesian Neural Networks (BNNs), which place priors on network weights, our evidential-based approach sets priors directly over the likelihood function. This methodology achieves a more computationally efficient uncertainty quantification.

We consider the uncertainty representations **z** extracted from the preceding autoencoder, which encapsulates the evidential information about normal network traffic, to conform to independent homogeneous distributions from a Gaussian distribution. These distributions are characterized by their mean and variance $(\mu, \sigma^2)$. These parameters to be quantified, $\mu$ and $\sigma^2$, are intrinsically linked to the model uncertainty that is the focus of our investigation [1]. To estimate these parameters, we employ a hierarchical Bayesian approach. Specifically, we utilize a Gaussian prior to estimate the mean value and place an Inverse-Gamma prior on the variance. This choice of priors is motivated by their conjugate relationship with the Gaussian likelihood, facilitating closed-form posterior updates. The hierarchical model can be expressed as follows:

$$\mathbf{z} \sim \mathcal{N}(\mu, \sigma^2) \qquad \mu \sim \mathcal{N}(\gamma, \sigma^2 v^{-1}) \qquad \sigma^2 \sim \Gamma^{-1}(\alpha, \beta), \quad (1)$$

where $\Gamma(\cdot)$ denotes the Gamma function, $\gamma$ represents the uncertainty space to be estimated, $v > 0$, $\alpha > 1$ and $\beta > 0$. We aim to

estimate a posterior distribution $q(\mu, \sigma^2 | \mathbf{z})$. Following the approach described in work [6], we employ a factorization of the estimated distribution such that $q(\mu, \sigma^2) = q(\mu)q(\sigma^2)$. This factorization allows for a tractable approximation of the posterior distribution. Our approximation takes the form of the Gaussian conjugate prior, specifically the Normal Inverse-Gamma (NIG) distribution:

$$p(\{\mu, \sigma^2\} \mid \Omega) = \frac{\beta^\alpha \sqrt{v}}{\Gamma(\alpha)\sqrt{2\pi\sigma^2}} \left(\frac{1}{\sigma^2}\right)^{\alpha+1} \exp\left\{-\frac{2\beta + v(\gamma-\mu)^2}{2\sigma^2}\right\},$$
(2)

where $\Omega = \{\gamma, v, \alpha, \beta\}$ denotes the set of uncertainty parameters we aim to estimate. Given a NIG distribution parameterized by $\Omega$, we can compute the uncertainty space and model uncertainty:

$$\underbrace{\mathbb{E}[\mu] = \gamma}_{\text{uncertainty space}} \qquad \underbrace{\text{Var}[\mu] = \frac{\beta}{v(\alpha-1)}}_{\text{model uncertainty}}.$$
(3)

This mathematical formulation delineates the theoretical framework underpinning our approach to uncertainty quantification. The evidential learning paradigm we have introduced essentially constitutes an uncertainty estimation methodology based on the likelihood function. This approach involves training a neural network to output the hyperparameters for fitting an evidential distribution.

Next, we outline our method for obtaining evidential parameters. Our training process is designed to optimize a dual-objective function that simultaneously addresses two critical aspects: (i) increasing model evidence to support the training samples, which in this context represent normal network traffic patterns, and (ii) reducing evidence when uncertainty space exhibits inconsistencies or inaccuracies. Objective (i) can be conceptualized as a mechanism for adapting our data to the evidential model, while objective (ii) serves to enforce a prior that mitigates inaccurate evidence and amplifies uncertainty where appropriate.

**Objective (i): Maximizing the Normal Evidence.** In accordance with Bayesian probability theory, the "model evidence" is defined as the likelihood of an observation, given the evidential distribution parameters $\Omega$. This is computed by marginalizing over the likelihood parameters $(\mu, \sigma^2)$:

$$p(\mathbf{z} \mid \Omega) = \int_{\sigma^2=0}^{\infty} \int_{\mu=-\infty}^{\infty} p\left(\mathbf{z} \mid \mu, \sigma^2\right) p\left(\mu, \sigma^2 \mid \Omega\right) \mathrm{d}\mu \mathrm{d}\sigma^2.$$
(4)

The direct fitting of the evidential model parameters $\Omega$ to this likelihood distribution presents significant computational challenges. However, by applying a Normal Inverse-Gamma (NIG) evidential prior to the Gaussian likelihood function, we can derive an analytical solution, as demonstrated in work [6]:

$$p(\mathbf{z} \mid \Omega) = \text{St}\left(\mathbf{z}; \gamma, \frac{\beta(1+v)}{v\alpha}, 2\alpha\right),$$
(5)

where $\text{St}(\cdot; \mu_{\text{St}}, \sigma_{\text{St}}^2, v_{\text{St}})$ denotes the Student's t-distribution evaluated at location parameter $\mu_{\text{St}}$, scale parameter $\sigma_{\text{St}}^2$, and degrees of freedom $v_{\text{St}}$. To optimize the model's representation of normal network traffic, we maximize the logarithm of the model evidence, which is equivalent to minimizing its negative. This objective guides the uncertainty parameter heads to output the parameters of a NIG distribution that best fits the distribution of normal network traffic.

Formally, we define the training objective $\mathcal{L}^{\text{NLL}}$ for maximizing the normal evidence as:

$$\mathcal{L}^{\text{NLL}} = \frac{1}{2}\log\left(\frac{\pi}{v}\right) - \alpha\log(\omega) + \log\left(\frac{\Gamma(\alpha)}{\Gamma\left(\alpha + \frac{1}{2}\right)}\right)$$
$$+ \left(\alpha + \frac{1}{2}\right)\log\left((\mathbf{z} - \gamma)^2 v + \omega\right),$$
(6)

where $\omega = 2\beta(1+v)$.

**Objective (ii): Minimizing Evidence on Errors.** In addition to maximizing the evidence for normal patterns, we incorporate a regularization term that imposes a high uncertainty prior to penalize incorrect evidence in the uncertainty space. The fundamental principle underlying this regularization is that it should attenuate the weight of evidence where the uncertainty space deviates significantly from the true evidence while having minimal impact on evidence predictions that closely align with the instructive evidence $\mathbf{z}$. To achieve this, we formulate an evidence regularizer [6] $\mathcal{L}^{\text{R}}$ as:

$$\mathcal{L}^{\text{R}} = |\mathbf{z} - \gamma| \cdot (2v + \alpha).$$
(7)

*3.3.3 Training.* Our training loss function comprises three principal components: $\mathcal{L}^{\text{NLL}}$, $\mathcal{L}^{\text{R}}$, and $\mathcal{L}^{\text{Rec}}$:

$$\mathcal{L} = \mathcal{L}^{\text{Rec}} \cdot \lambda_{\text{Rec}} + \mathcal{L}^{\text{NLL}} \cdot \lambda_{\text{NLL}} + \mathcal{L}^{\text{R}} \cdot \lambda_{\text{R}},$$
(8)

where $\lambda.$ is the hyperparameter to control the contribution of each component. $\mathcal{L}^{\text{Rec}}$ is the reconstruction loss for the autoencoder:.

$$\mathcal{L}^{\text{Rec}} = ||\mathbf{x} - \hat{\mathbf{x}}||_1,$$
(9)

where $||\cdot||_1$ denotes the L1 norm. The inclusion of this term ensures the preservation of the autoencoder's fundamental reconstruction capability, enabling the generation of meaningful latent representations. These representations serve as effective evidence instructors for the subsequent uncertainty quantification.

*3.3.4 Inference.* The anomaly detection process fundamentally relies on an anomaly score to quantify the degree of deviation from normality. Given that our model is trained exclusively on normal network traffic, the proposed UnDiff naturally assigns low uncertainty to patterns consistent with normal network behavior. Our approach is motivated by the well-established principle that there exists a distributional divergence between normal and anomalous network traffic, encompassing both undrifted and drifted anomalies [8, 19, 51]. Leveraging this insight, we adopt an inter-sample differences method, utilizing model uncertainty as a direct proxy for anomaly scoring. This approach is underpinned by the widely accepted notion in uncertainty learning that deviant samples inherently induce higher model uncertainty [20]. As depicted in Figure 2(b), our method yields an effective and computationally efficient uncertainty-inspired anomaly score. This score is characterized by its ability to generate high uncertainty values for anomalous samples (i.e., out-of-distribution instances relative to the training set) while maintaining low uncertainty for normal samples (i.e., in-distribution instances relative to the training set). In contrast to traditional reconstruction-based anomaly quantification methods, which we categorize as intra-sample difference approaches, UnDiff capitalizes on the intrinsic distributional divergence between normal and anomalous network traffic. This enables a more nuanced

Table 2: Performance comparisons (%) for undrifted anomaly detection on the DataCon2020, CIC-IDS2017, and USTC-TFC2016 datasets. The best results are in bold, and the runner-up results are underlined.

| Model | DataCon2020 | | | CIC-IDS2017 | | | USTC-TFC2016 | | |
|---|---|---|---|---|---|---|---|---|---|
| | AUC | ACC | F1 | AUC | ACC | F1 | AUC | ACC | F1 |
| PaDim | 61.01±0.5 | 55.07±0.2 | 67.23±0.2 | 57.74±0.4 | 55.06±0.1 | 68.16±0.1 | 98.79±0.0 | 96.93±0.1 | 96.85±0.1 |
| DFM | 83.03±0.3 | 78.67±0.3 | 78.85±0.3 | 69.91±0.2 | 63.89±0.3 | 66.87±0.1 | 94.94±0.3 | 93.03±0.3 | 92.63±0.3 |
| DFKDE | 72.85±0.5 | 64.37±0.3 | 71.86±0.2 | 67.67±0.6 | 63.62±0.3 | 68.61±0.2 | 91.63±2.4 | 93.38±0.2 | 93.79±0.2 |
| Fastflow | 69.98±0.5 | 63.93±0.3 | 71.65±0.3 | 78.25±0.3 | 74.82±0.4 | 76.39±0.4 | 99.14±0.0 | 95.60±0.2 | 95.42±0.2 |
| Cflow | 68.69±1.6 | 64.14±1.0 | 72.40±0.8 | 66.42±0.9 | 69.46±0.6 | 69.41±0.5 | 97.22±0.2 | 96.76±0.2 | 96.68±0.2 |
| STFPM | 82.37±0.6 | 80.44±2.1 | 80.93±1.7 | 85.89±1.7 | 80.00±2.1 | 81.50±1.1 | 91.63±2.4 | 89.02±1.3 | 89.71±1.3 |
| ReverDis | 74.53±2.4 | 68.80±3.1 | 75.30±1.1 | 82.22±0.3 | 77.87±0.4 | 76.62±0.4 | 98.05±0.5 | 95.21±0.9 | 95.07±0.9 |
| MMR | 80.60±2.1 | 78.85±2.1 | 79.64±1.3 | 85.87±1.2 | 74.36±1.1 | 74.40±1.0 | 99.44±0.0 | 96.15±0.2 | 96.04±0.2 |
| GANomaly | 81.50±1.0 | 79.40±2.1 | 79.95±1.6 | 82.75±4.7 | 80.85±1.7 | 81.21±0.9 | 95.36±1.0 | 91.27±2.9 | 91.07±3.2 |
| ARCADE | 81.98±4.1 | 81.48±2.0 | 80.31±3.4 | 84.85±2.6 | 80.15±1.6 | 82.78±1.0 | 88.62±2.2 | 93.13±0.1 | 93.57±0.1 |
| MFAD | 83.16±1.9 | 76.28±2.4 | 78.59±1.1 | 86.02±0.8 | 81.66±1.9 | 83.67±1.7 | 99.73±0.0 | 97.45±0.4 | 97.43±0.4 |
| Trident | 63.89±0.5 | 73.67±0.3 | 78.37±0.3 | 82.99±0.1 | 77.42±0.2 | 75.17±0.2 | 96.19±0.2 | 89.86±0.3 | 89.47±0.3 |
| **UnDiff (ours)** | **86.93±0.3** | **83.16±0.2** | **82.78±0.2** | **88.88±0.4** | **83.31±0.4** | **83.72±0.4** | **99.90±0.0** | **99.47±0.2** | **99.47±0.2** |
| Improve | 4.53%↑ | 2.06%↑ | 2.29%↑ | 3.32%↑ | 2.02%↑ | 0.18%↑ | 0.17%↑ | 0.02%↑ | 0.04%↑ |

and potentially more robust detection mechanism. Formally, we define our anomaly score as follows:

$$\text{Anomaly Score} = \text{Var}[\mu] = \frac{\beta}{v(\alpha - 1)}. \tag{10}$$

## 4 Experiments

In this section, we present a comprehensive empirical evaluation to assess the efficacy of UnDiff. Our experiments aim to answer the following research questions, each probing a critical aspect of our proposed model:

- **RQ1:** To what extent does UnDiff demonstrate superior performance relative to established baselines in detecting undrifted anomalies across multiple datasets?
- **RQ2:** How does the efficacy of UnDiff compare to baselines when confronted with drifted anomalies?
- **RQ3:** What is the relative impact of intra- and inter-sample differences on the detection performance?
- **RQ4:** What is the contribution of the various components within UnDiff to its overall detection capability?

### 4.1 Experimental Setting

**Dataset.** We use three publicly available network traffic anomaly detection datasets for evaluation: (i) *DataCon2020* [9] is an encrypted network traffic dataset comprising normal and malicious traffic, with the latter consisting of encrypted malware communications; (ii) *CIC-IDS2017* [35] is a network intrusion detection dataset that includes seven common attacks, including Brute Force Attack, Heartbleed Attack, Botnet, DoS, DDoS, Web Attack, and Infiltration Attack; (iii) *USTC-TFC2016* [39] is malware traffic detection dataset with malicious traffic from public sources and normal traffic from eight application types. For consistent evaluation, we randomly sample 10,000 normal network flows for training and 5,000 normal plus 5,000 anomalous flows for testing across all datasets.

**Baselines.** We evaluate UnDiff with 12 state-of-the-art baselines, categorized into two groups as follows: (i) *Network Traffic Anomaly Detection*: *GANomaly* [4], *ARCADE* [31], *MFAD* [51], and *Trident* [50]; (ii) *Other Advanced Anomaly Detection*: *PaDim* [10], *DFM*

[2], *DFKDE* [3], *FastFlow* [44], *CFlow* [17], *STFPM* [37], *ReverDis* [11], and *MMR* [48]. More details of baselines are shown in Appendix B.

**Evaluation Metrics.** In alignment with recent models in network traffic anomaly detection [31, 51], we employ three commonly used metrics: AUC, Accuracy (ACC), and F1-Score (F1).

**Drifted Anomaly.** We assess model's robustness to anomaly drift by conducting cross-dataset evaluations. Specifically, we train one model on one dataset and evaluate this model's performance on anomalous samples from other two datasets. This approach allows us to investigate model's generalization capability and resilience to potential concept shifts in network traffic patterns, thereby assessing model's efficacy in detecting drifted anomalies in real-world environments.

**Implementation Details.** All experiments are conducted on an NVIDIA GeForce RTX 3090 GPU. We use the Adam optimizer with learning rates of $1e^{-4}$, $1e^{-3}$, $1e^{-6}$ for DataCon2020, CIC-IDS2017, and USTC-TFC2016, respectively. Loss coefficient ($\lambda_{\text{Rec}}$, $\lambda_{\text{NLL}}$, $\lambda_{\text{R}}$) are set as $(1, 1e^{-2}, 1e^{-4})$, $(1, 5e^{-2}, 5e^{-5})$ and $(1, 1, 1e^{-2})$, while the low-pass filter uses a cutoff radius of 5. The parameter search scope is described in Appendix C. Training proceeds with a batch size of 128 for a maximum of 50 epochs, with early stopping implemented to mitigate overfitting. To ensure statistical robustness, we perform five independent runs with different random seeds, reporting mean results with standard deviations. To facilitate reproducibility, the source code for our UnDiff is available at https://anonymous.4open.science/r/WWW25-1522 and will be made public.

### 4.2 Anomaly Detection on Benchmark (RQ1)

To assess UnDiff's efficacy in typical anomaly traffic detection scenarios (i.e., undrifted anomalies), we conducted a comprehensive comparison of our model against 12 competitive baselines on three datasets. The results, as presented in Table 2, demonstrate that our model consistently outperforms baselines across all three datasets. Notably, on the DataCon2020 and CIC-IDS2017 datasets, UnDiff exhibits significant performance improvements over the best-performing baseline MFAD, with enhancements of 4.53%↑ and 3.32%↑ in AUC, respectively. These quantitative improvements

**Table 3: Performance comparisons (%) for drifted anomaly detection on the DataCon2020, CIC-IDS2017, and USTC-TFC2016 datasets. The abbreviations are explained as follows: D: DataCon2020, I: CIC-IDS2017, and U: USTC-TFC2016.**

| Model | D->I | | | D->U | | | I->D | | | I->U | | | U->D | | | U->I | | |
|---|---|---|---|---|---|---|---|---|---|---|---|---|---|---|---|---|---|---|
| | AUC | ACC | F1 | AUC | ACC | F1 | AUC | ACC | F1 | AUC | ACC | F1 | AUC | ACC | F1 | AUC | ACC | F1 |
| GANomaly | 50.73 | 50.99 | 66.86 | 60.96 | 75.82 | 79.68 | 56.58 | 72.35 | 75.76 | 74.92 | 72.32 | 77.49 | 90.30 | 90.70 | 91.25 | 96.01 | 88.67 | 87.95 |
| ARCADE | 49.99 | 49.99 | 66.66 | 57.87 | 74.83 | 78.74 | 67.47 | 79.24 | 80.94 | 49.60 | 59.74 | 69.60 | 97.87 | 93.44 | 93.84 | 88.61 | 90.67 | 91.44 |
| MFAD | 62.28 | 51.09 | 67.01 | 79.50 | 72.76 | 77.90 | 77.90 | 70.56 | 74.60 | 82.18 | 75.84 | 78.02 | 98.20 | 95.27 | 95.43 | 98.62 | 93.92 | 94.19 |
| Trident | 55.11 | 49.90 | 66.66 | 66.08 | 78.99 | 81.77 | 51.20 | 57.90 | 70.07 | 67.47 | 66.24 | 72.57 | 97.47 | 98.28 | 98.31 | 98.83 | 97.57 | 97.62 |
| UnDiff-AE | 64.48 | 70.57 | 77.10 | 71.08 | 76.97 | 80.73 | 81.48 | 79.21 | 80.47 | 61.88 | 62.27 | 71.91 | 99.57 | 98.92 | 98.92 | 98.75 | 92.80 | 93.27 |
| **UnDiff** | **84.10** | **76.14** | **80.46** | **96.08** | **87.95** | **88.66** | **93.70** | **88.65** | **88.57** | **91.18** | **86.43** | **86.69** | **99.83** | **99.59** | **99.59** | **99.76** | **98.07** | **98.07** |
| Improve | 30.43%↑ | 7.89%↑ | 4.36%↑ | 20.86%↑ | 11.34%↑ | 8.43%↑ | 15.00%↑ | 11.88%↑ | 9.43%↑ | 10.95%↑ | 10.59%↑ | 13.96%↑ | 0.26%↑ | 0.68%↑ | 0.68%↑ | 0.94%↑ | 0.51%↑ | 0.46%↑ |

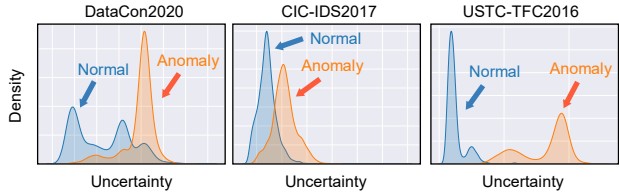

**Figure 3: Statistics of the uncertainty-based anomaly scores for UnDiff under the undrifted anomaly scenario.**

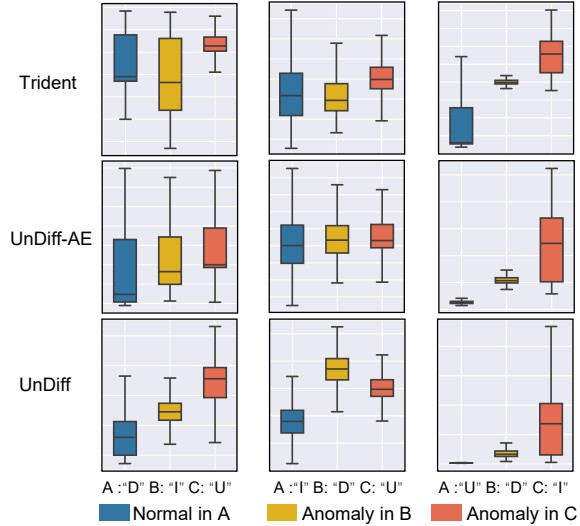

**Figure 4: Anomaly score distribution for Trident, UnDiff-AE, and UnDiff under the anomaly drift scenario.**

underscore the effectiveness and robustness of our model. The underlying strength of our method lies in its innovative utilization of uncertainty measures to directly quantify inter-sample differences, thereby facilitating more accurate discrimination of anomalous network traffic patterns.

While statistics-based methods such as PaDim, DFM, and DFKDE, as well as normalizing flow-based approaches like FastFlow and CFlow, attempt to compute distribution deviations by exploiting inter-sample differences, their comparative spaces lack the discriminative power of our uncertainty space. Our approach, built upon the informative reconstruction of latent variables and guided by evidential learning, constructs a more robust and discerning comparative framework. Moreover, distillation-based methods in anomaly detection, including STFPM, ReverDis, and MMR, are constrained by the limitation in effective feature extraction. In contrast, approaches specific to anomaly network traffic detection, while not requiring additional feature extractors, quantify anomalies through intra-sample reconstruction differences. However, these methods, including GANomaly, ARCADE, MFAD, and Trident, suffer from the "identical shortcut" issue, which may significantly compromise the intra-sample differences of anomalies, leading to suboptimal performance. Our uncertainty-inspired framework addresses these limitations by effectively leveraging distributional differences between normal and anomalous samples. By quantifying anomalies from an inter-sample differences perspective, UnDiff provides a more nuanced and robust approach to anomaly detection.

To further corroborate the feasibility of our UnDiff framework, we present a detailed analysis of the anomaly score distributions in Figure 3. The graphical representation reveals a marked bimodal distribution, with a clear separation between the scores associated with normal and anomalous samples. This pronounced divergence in score distributions provides compelling evidence for the discriminative power of our uncertainty-inspired anomaly metric. The clear detachment between normal and anomalous samples also

underscores the method's ability to generate highly informative indicators, facilitating more accurate and reliable anomaly detection.

## 4.3 Drifted Anomaly Detection (RQ2)

To assess the efficacy of our approach in addressing drifted anomalies, we compare our UnDiff with state-of-the-art network traffic anomaly detection methods and a variant of our UnDiff – UnDiff-AE, which employs a pure auto-encoder architecture without uncertainty learning. As evidenced in Table 3, these approaches have suboptimal performance, particularly in the drifted experiments from DataCon2020 (D) to CIC-IDS2017 (I), D to USTC-TFC2016 (U), I to D, and I to U. These empirical observations highlight the critical necessity for robust drifted anomaly detection methodologies. The primary limitation of these baselines stems from their reliance on an intra-sample difference paradigm, which is inherently susceptible to the "identical shortcut" issue prevalent in reconstruction-based models. Therefore, the divergence in anomaly scores between normal and anomalous samples is suppressed and obfuscated. We visualize the detailed anomaly scores for Trident and UnDiff-AE in Figure 4 to elucidate this phenomenon. The anomaly score distribution for Trident exhibits significant overlap between normal and anomalous samples, with anomalous samples occasionally scoring lower than normal samples. This observation indicates that the "identical shortcut" issue profoundly compromises the efficacy of

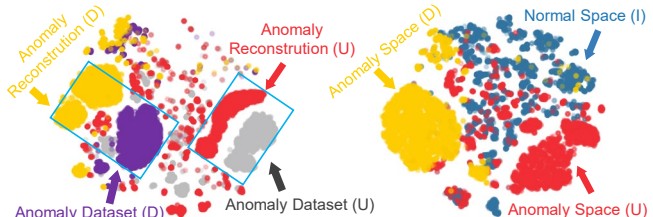

Trident: Intra-Sample Differences    UnDiff: Inter-Sample Differences

**Figure 5: The t-SNE visualization comparison between intra-sample and inter-sample differences.**

intra-sample differences in detecting drifted anomalies. In contrast, UnDiff achieves apparent distinction in the distribution between normal and drifted anomaly samples, thereby validating the effectiveness of our inter-sample differences approach. Notably, the comparative analysis with UnDiff-AE shows that the substantial improvement in UnDiff's performance is predominantly attributable to the uncertainty-inspired inter-sample differences rather than the fundamental auto-encoder architecture. Appendix D provides more comparison results.

### 4.4  Qualitative Study (RQ3)

We now elucidate the underlying mechanisms contributing to UnDiff's enhanced performance by two *t*-SNE visualizations, which leverage an inter-sample differences methodology. As illustrated in Figure 5, we observe a degree of confusion between the pre- and post-reconstruction embeddings of anomalous samples on Trident, manifested as certain overlaps and approximate profiles with minimal distance. We posit that this phenomenon arises from the "identical shortcut" issue, an inherent limitation in reconstruction-based approaches. This limitation leads to well-reconstructed representations even for anomalous samples, a phenomenon that contradicts the fundamental detection motivation of reconstruction methods. Consequently, this results in indistinguishable intra-sample differences between normal and anomalous traffic patterns, compromising the efficacy of traditional approaches. In contrast, UnDiff is based on a novel inter-sample differences perspective, effectively leveraging the axiom that anomalous samples inherently deviate from normal samples in the feature space. The representation within our uncertainty space demonstrates the feasibility and effectiveness of uncertainty-inspired modeling and detection. This approach makes the discrimination by exploiting inter-sample differences, thereby overcoming the limitations inherent in intra-sample comparison methods. More qualitative results are in Appendix E.

### 4.5  Ablation Studies (RQ4)

To evaluate the contributions of each component in our UnDiff, we conduct an ablation study comprising four variants. These variants are constructed by removing one of the key components in UnDiff: the reconstruction loss $\mathcal{L}^{\text{Rec}}$ (w/o $\mathcal{L}^{\text{Rec}}$), the regularization loss $\mathcal{L}^{\text{R}}$ (w/o $\mathcal{L}^{\text{R}}$), the uncertainty-based anomaly score (w/o AS), and both the uncertainty-based modeling and anomaly score (w/o T&AS). As illustrated in Table 4, the removal of $\mathcal{L}^{\text{Rec}}$ results in substantial performance degradation, underscoring the critical role of reconstruction loss in ensuring a refined representation of normal network

**Table 4: Ablation studies for drifted and undrifted anomaly detection (AUC). The gray color denotes undrifted detection.**

| Variant | DataCon2020 | | | CIC-IDS2017 | | | USTC-TFC2016 | | |
|---|---|---|---|---|---|---|---|---|---|
| | D | I | U | D | I | U | D | I | U |
| w/o $\mathcal{L}^{\text{Rec}}$ | 83.44 | 61.79 | 80.26 | 78.46 | 76.22 | 59.41 | 98.95 | 80.97 | 98.25 |
| w/o $\mathcal{L}^{\text{R}}$ | 84.80 | 71.48 | 72.50 | 85.31 | 86.80 | 81.12 | 99.55 | 99.39 | 99.78 |
| w/o AS | 85.66 | 82.52 | 83.25 | 83.28 | 87.67 | 86.15 | 99.74 | 98.81 | 99.84 |
| w/o T&AS | 85.55 | 64.48 | 71.08 | 81.48 | 86.02 | 61.88 | 99.51 | 98.75 | 99.71 |
| **UnDiff** | **86.93** | **84.10** | **96.08** | **93.70** | **88.88** | **91.18** | **99.83** | **99.76** | **99.90** |

**Table 5: Overhead comparison for inference.**

| | GANomaly | ARCADE | MFAD | Trident | UnDiff |
|---|---|---|---|---|---|
| MACs (G) | 0.98 | 0.82 | 0.99 | **0.03** | 0.25 |
| #Paras (M) | 9.66 | 6.7 | 10.07 | 27.61 | **2.55** |

traffic. Furthermore, we observe a notable decline in performance upon removal of $\mathcal{L}^{\text{R}}$, indicating its efficacy as a regularization constraint in preventing the formation of erroneous evidence spaces during the uncertainty quantification process. While removing the uncertainty-based anomaly score (w/o AS) and both uncertainty-based modeling and anomaly score (w/o T&AS) resulted in performance degradation, our complete UnDiff model demonstrates optimal performance in drifted anomaly detection. This suggests that the uncertainty-based modeling and inter-sample difference detection components effectively leverage prior differences between normal and anomalous samples, mitigating the inherent limitations of purely reconstruction-based methods. More experimental results using other metrics are presented in the Appendix F.

### 4.6  Overhead evaluation

We conduct an analysis of model efficiency, focusing on multiply-accumulate operations per second (MACs) and the number of model parameters (#Paras) during inference. The results of this analysis are summarized in Table 5. Our UnDiff demonstrates excellent performance with favorable computational overhead compared to alternative baselines. This efficiency can be attributed to strategic design choices, such as the multi-channel image representation, a low-parameter evidence extractor, and a set of concise uncertainty parameter heads. Notably, UnDiff balances performance and computational requirements, rendering it particularly suitable for practical deployment in network traffic anomaly detection scenarios.

## 5  Conclusions

This study presents a pioneering approach to network traffic anomaly detection by developing an inter-sample differences method based on uncertainty. This novel methodology directly addresses the challenges of anomaly detection while circumventing the "identical shortcut" issue inherent in existing methods that rely on intra-sample differences between pre- and post-reconstruction representations. Our proposed UnDiff effectively leverages the prior knowledge that anomalous samples inherently deviate from normal samples. This enables learning a more discriminative uncertainty space, facilitating optimal detection performance. Comprehensive empirical evaluations across three benchmark datasets demonstrate UnDiff's superior performance in detecting undrifted and drifted anomalies with minimal additional computational overhead.

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

## A  Additional Details for Model Design

UnDiff consists of two main components: the autoencoder and the uncertainty learning module. The autoencoder follows a standard architecture comprising 4 convolutional layers and 4 deconvolutional layers. Each convolutional and deconvolutional layer is followed by a Convolutional Block Attention Module (CBAM) [40], which allows the model to prioritize significant channel images (temporal features) and spatially relevant regions (spatial features). This design aids in extracting key evidence from critical patterns in multi-channel images. The detailed number of channels in the convolutional layers is [32, 64, 128, 256], with a kernel size of 3 for all layers.

## B  Baselines

(1) *Advanced anomaly detection methods*, including *PaDim* [10], *DFM* [2], *DFKDE* [3], *FastFlow* [44], *CFlow* [17], *STFPM* [37], *ReverDis* [11], and *MMR* [48] are implemented with official code. Notably, our multi-channel image data type may not satisfy their needs. As a solution, we concatenate traffic images with spatial dimension instead of channel dimension to incorporate these methods. (2) *Network traffic anomaly detection methods*, such as *GANomaly* [4] and *Trident* [50], are also implemented with official code. For *AR-CADE* [31] and *MFAD* [51], we reproduce them from the original paper. To ensure a fair comparison, we perform the grid search for their training parameters and learning rate, setting the search range according to the original articles for each baseline. We provide implementation details of network traffic anomaly detection methods as follows:

- GANomaly [4]. GANomaly is a classical comparison method used for anomaly detection in network traffic, featuring an encoder-decoder-encoder architecture. Since it does not consider the effects of image filtering, we use unfiltered images as input. The network architecture is based on the official implementation, and hyper-parameters are adjusted to better adapt to the distributions of different traffic datasets.
- ARCADE [31]. To process image data inputs like other baselines, we use a 2D convolutional layer to build the encoder-decoder architecture. The WGAN-GP strategy is incorporated into the framework, and hyper-parameters are adjusted according to the specific traffic datasets.
- MFAD [51]. MFAD is an ensemble-based method utilizing multi-scale filtering. Since the fusion coefficients need to be determined based on ground truth labels, we only use the best-performing low-frequency branch for the final results. The filtering methods and scales follow the original paper, and hyper-parameters are adjusted for different traffic datasets.
- Trident [4]. We follow the official code for implementation, and the learning rate is fine-tuned to optimize performance on different datasets.
- UnDiff-AE. This variant of our UnDiff method serves to demonstrate the effectiveness of our approach. In UnDiff-AE, we remove the uncertainty parameter heads from the uncertainty module but retain the encoder-decoder-encoder architecture. It trains using reconstruction loss and uses

standard reconstruction error to quantify anomalies, providing a comparison with the full UnDiff model.

## C  Complete Implementation Details

We use the Adam optimizer with a weight decay of 0.05 and a batch size of 128. The maximum training epoch is 50, and early stopping is applied to prevent overfitting. The training process will halt when parameter updates no longer yield improvements for 6 epochs. For the three datasets, we perform a grid search for learning rates within the range $(1e^{-2}, 1e^{-3}, 1e^{-4}, 1e^{-5}, 1e^{-6})$, and the final learning rates are set to $1e^{-4}$, $1e^{-3}$ and $1e^{-6}$ respectively. The search scope of low-pass filtering is (15, 10, 5), and for the reconstruction loss coefficient $\lambda_{Rec}$ is $(1, 5e^{-1}, 1e^{-1}, 5e^{-2}, 1e^{-2})$. Similarly, the search scope for loss coefficients $\lambda_{NLL}$ and $\lambda_R$ is $(1, 5e^{-1}, 1e^{-1}, ..., 5e^{-3}, 1e^{-3})$ and $(1, 5e^{-1}, 1e^{-1}, ..., 5e^{-5}, 1e^{-5})$, respectively. The final loss coefficients $(\lambda_{Rec}, \lambda_{NLL}, \lambda_R)$ are set to $(1, 1e^{-2}, 1e^{-4})$, $(1, 5e^{-2}, 5e^{-5})$ and $(1, 1, 1e^{-2})$. A relatively larger $\lambda_{Rec}$ ensures stable autoencoder training, making it an effective evidence extractor. In contrast, $\lambda_R$ is a penalty term to assist in modeling evidence. Therefore, it is kept low to avoid overwhelming the main evidence modeling loss, which could lead to training instability.

## D  Complete Drifted Anomaly Detection

Figure 6 illustrates the detailed anomaly score distribution for all anomaly detection scenarios in network traffic, including normal samples (light blue distribution), undrifted anomalies (dark blue distribution), and other drifted anomalies (red and orange distributions) across three datasets. The baseline results show significant confusion between normal and anomalous scores, particularly in the DataCon2020 (D) and CIC-IDS2017 (I) datasets. This highlights the insufficient discriminative ability of reconstruction-based methods, which rely on an intra-sample differences paradigm and are inherently prone to the "identical shortcut" issue common in such models. As a result, the divergence in anomaly scores between normal and anomalous samples is suppressed and often obfuscated.

Notably, in scenarios such as DataCon2020 (D) to CIC-IDS2017 (I), a counter-intuitive phenomenon occurs where drifted anomalies exhibit more minor deviations than normal samples, further exposing the limitations of reconstruction models in handling such cases. In contrast, our UnDiff method effectively leverages inter-sample differences (i.e., the inherent distributional divergence between normal and anomalous samples), resulting in the most distinct separation between normal samples (light blue) and anomalies (other colors).

## E  Complete Qualitative Study

As shown in Figure 7, we provide a complete t-distributed stochastic neighbor embedding (t-SNE) comparison between Trident and our UnDiff method. In the top row of Figure 7, a common pattern emerges where pre- and post-reconstruction embeddings of anomalous samples show a degree of confusion, marked by embedding overlaps and closely aligned profiles (highlighted in the blue box). This indicates the potential presence of the "identical shortcut" issue. For UnDiff, which employs an inter-sample differences-based quantification, we visualize the first layer of the uncertainty parameters.

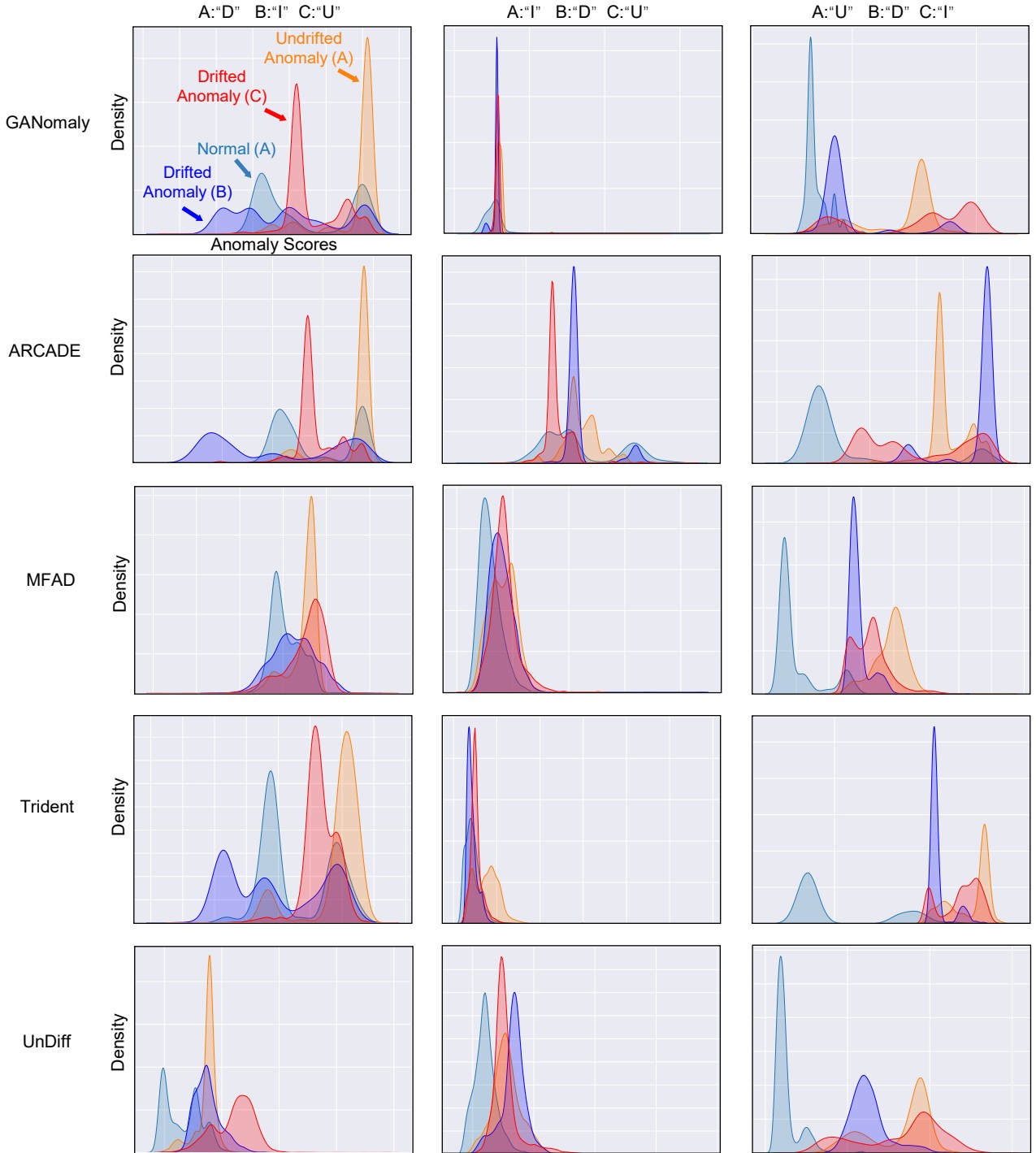

**Figure 6: Complete anomaly score distribution comparing other methods with our UnDiff, including both drifted and undrifted detection performance. The abbreviations are explained as follows: D: DataCon2020, I: CIC-IDS2017 and U: USTC-TFC2016.**

As seen in the bottom row of Figure 7, the distributional embeddings for normal and drifted anomalies form distinct clusters. This demonstrates that our designed uncertainty module effectively generates an uncertainty distribution in the uncertainty space, where normal samples exhibit low uncertainty and anomalous samples show high uncertainty. By leveraging this distinct representation,

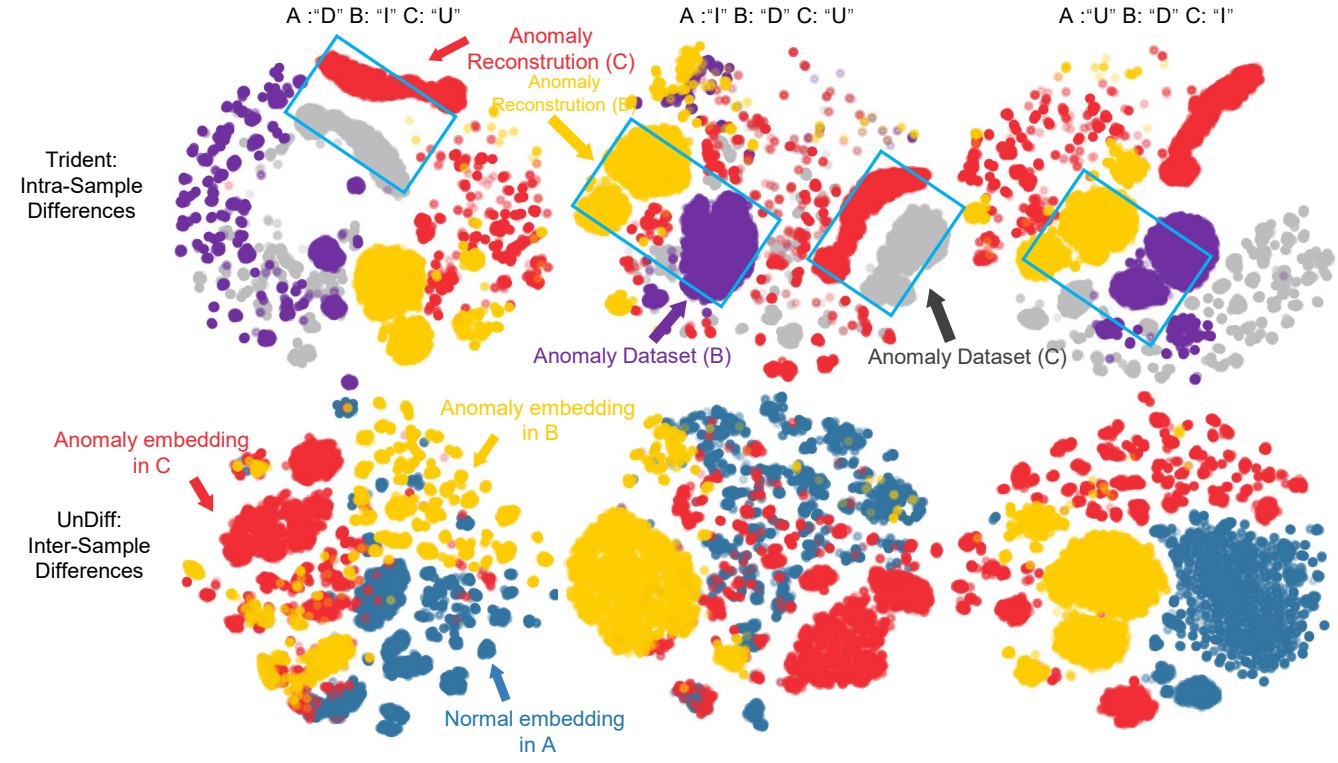

**Figure 7: Complete t-SNE visualization results comparing the Trident (intra-sample differences) with our UnDiff (inter-sample differences) method.**

**Table 6: Ablation studies (%) for drifted anomaly detection on the DataCon2020, CIC-IDS2017 and USTC-TFC2016 datasets. The abbreviations are explained as follows, D: DataCon2020, I: CIC-IDS2017 and U: USTC-TFC2016.**

| Variant | D->I | | | D->U | | | I->D | | | I->U | | | U->D | | | U->I | | |
|---|---|---|---|---|---|---|---|---|---|---|---|---|---|---|---|---|---|---|
| | AUC | ACC | F1 | AUC | ACC | F1 | AUC | ACC | F1 | AUC | ACC | F1 | AUC | ACC | F1 | AUC | ACC | F1 |
| w/o LPF | 74.73 | 69.96 | 76.41 | 68.75 | 68.81 | 75.02 | 70.57 | 80.08 | 82.04 | 70.07 | 70.17 | 76.2 | 98.21 | 96.18 | 96.25 | 98.04 | 93.14 | 93.58 |
| w/o $\mathcal{L}^{Rec}$ | 61.79 | 60.18 | 69.96 | 80.26 | 76.24 | 80.24 | 78.46 | 76.66 | 79.49 | 59.41 | 64.05 | 71.73 | 98.95 | 96.71 | 96.71 | 80.97 | 86.72 | 85.75 |
| w/o $\mathcal{L}^R$ | 71.48 | 67.58 | 74.22 | 72.50 | 77.06 | 78.33 | 85.31 | 82.95 | 82.46 | 81.12 | 80.05 | 80.31 | 99.55 | 98.97 | 98.97 | 99.39 | 95.67 | 95.79 |
| w AS-trans | 57.95 | 51.85 | 67.44 | 82.20 | 83.53 | 84.24 | 81.00 | 74.80 | 77.69 | 80.96 | 79.68 | 80.37 | 99.5 | 98.94 | 98.95 | 96.16 | 93.03 | 93.48 |
| w/o AS | 82.52 | 74.21 | 79.84 | 83.25 | 85.66 | 86.68 | 83.28 | 80.46 | 82.08 | 86.15 | 80.44 | 79.95 | 99.74 | 99.32 | 99.32 | 98.81 | 97.73 | 97.73 |
| w/o T&AS | 64.48 | 70.57 | 77.10 | 71.08 | 76.97 | 79.73 | 81.48 | 79.21 | 80.47 | 61.88 | 62.27 | 71.91 | 99.51 | 98.92 | 98.92 | 98.75 | 92.80 | 93.27 |
| **UnDiff** | **84.10** | **76.14** | **80.46** | **96.08** | **87.95** | **88.66** | **93.70** | **88.65** | **88.57** | **91.18** | **86.43** | **86.69** | **99.83** | **99.59** | **99.59** | **99.76** | **98.07** | **98.07** |

**Table 7: Ablation studies (%) for undrifted anomaly detection on the DataCon2020, CIC-IDS2017 and USTC-TFC2016 datasets.**

| Variant | DataCon2020 | | | CIC-IDS2017 | | | USTC-TFC2016 | | |
|---|---|---|---|---|---|---|---|---|---|
| | AUC | ACC | F1 | AUC | ACC | F1 | AUC | ACC | F1 |
| w/o LPF | 81.03 | 81.38 | 81.74 | 81.27 | 76.93 | 79.47 | 99.76 | 98.22 | 98.23 |
| w/o $\mathcal{L}^{Rec}$ | 83.44 | 79.23 | 79.43 | 76.22 | 69.57 | 72.80 | 98.25 | 96.81 | 96.78 |
| w/o $\mathcal{L}^R$ | 84.80 | 82.14 | 82.80 | 86.80 | 83.06 | 83.22 | 99.78 | 98.57 | 98.57 |
| w AS-Trans | 84.77 | 83.00 | 82.01 | 85.90 | 75.65 | 78.62 | 99.40 | 97.64 | 97.67 |
| w/o AS | 85.66 | 83.01 | 82.66 | 87.67 | 82.69 | 84.22 | 99.84 | 99.33 | 99.33 |
| w/o T&AS | 85.55 | 82.76 | 82.87 | 86.02 | 79.49 | 81.59 | 99.71 | 98.40 | 98.39 |
| **UnDiff** | **86.93** | **83.16** | **82.78** | **88.88** | **83.31** | **83.72** | **99.9** | **99.47** | **99.47** |

our anomaly scores, based on inter-sample differences, fully utilize the prior distributional knowledge, significantly enhancing detection performance.

# F   Complete Ablation Studies

We conducted extensive ablation studies, with the complete results shown in Table 6 (drifted anomaly detection) and Table 7 (undrifted anomaly detection). Apart from removing the reconstruction loss $\mathcal{L}^{Rec}$ (w/o $\mathcal{L}^{Rec}$), the regularization loss $\mathcal{L}^R$ (w/o $\mathcal{L}^R$), the uncertainty-based anomaly score (w/o AS), and both the uncertainty-based modeling and anomaly score (w/o T&AS), we also report performance after removing the low-pass filtering (w/o LPF) and using a variant that trains with the uncertainty score but tests using the reconstruction loss (w AS-Trans).

Overall, we can observe a notable decline in performance upon removal of (w/o LPF) in both drifted and undrifted studies, indicating that the high-frequency information in traffic images hinders the uncertainty quantification process. The removal of (w/o $\mathcal{L}^{Rec}$) results in a significant decline in undrifted studies, highlighting its crucial role in extracting meaningful evidence and preventing the collapse of uncertainty modeling.

While the variant (w AS-Trans) shows a performance decline, it still achieves acceptable results under suboptimal testing conditions. This demonstrates UnDiff's robustness in uncertainty-based modeling, even when faced with suboptimal evaluation methods.

Comparing these results with the intra-sample differences limitations in reconstruction-based methods, we observe a substantial decline in performance when removing the uncertainty-based anomaly score (w/o AS) and(w/o T&AS). These findings suggest that the uncertainty-based modeling and inter-sample difference detection components effectively leverage prior knowledge of differences between normal and anomalous samples, mitigating the inherent limitations of purely reconstruction-based methods.

