# OpenReview forum: "Facing Anomalies Head-On: Network Traffic Anomaly Detection via Uncertainty-Inspired Inter-Sample Differences"
_ACM.org/TheWebConf/2025/Conference — WWW 2025 Poster_

### Official Review · Reviewer_CB41 · 2024-11-01

**Novelty:** 4
**Technical Quality:** 3

**Review:**

Summary：
	To address the poor detection performance of reconstruction-based network anomaly detection methods, particularly their difficulty in handling anomaly-drift issues, this paper proposes UnDiff, an uncertainty-inspired anomaly detection method. UnDiff determines the degree of anomaly through uncertainty deviation between samples, improving detection accuracy. Furthermore, extensive experiments validate the detection performance of UnDiff, showing its outstanding results, especially in anomaly-drift scenarios.

Strength：

1.	Introducing the relatively new machine learning method of Evidential Deep Learning into the field of network security is insightful.

2.	The illustrations and visualizations in the paper are good—very clear and concise.

3.	The experimental design is comprehensive, effectively demonstrating the performance of the proposed method.

Weakness:

1.	The writing still needs improvement, as it does not clearly articulate the motivation of the paper, and the insights behind the proposed method are somewhat vague. These issues are primarily evident in the introduction section:

1.1 The results on the right side of Figure 1 need to quantitatively demonstrate that the representation distances before and after reconstruction are very close; visualization alone may not effectively convey this issue.

1.2 The motivation for studying the anomaly drift problem needs to be emphasized. It is important to explain whether (or why) it poses a serious threat in real networks and why it is challenging to defend against, as this helps readers understand the contributions of this paper.

1.3 In the introduction section, a brief introduction to model uncertainty should be provided, as introducing this concept abruptly in the fourth paragraph may make it difficult for readers to comprehend.

1.4 This is just a suggestion: perhaps adding subtitles like "Limitations" and "Contributions" in the introduction is not necessary, as this may disrupt the flow of reading. Even without these subtitles, readers can still grasp the main ideas of that section.

2.	Regarding Novelty:

2.1 I greatly appreciate the authors' introduction of certain concepts from Evidential Deep Learning to network anomaly detection, which is very inspiring. However, I do not believe that the focus on anomaly drift is a challenging issue. Anomaly drift itself may not pose a detection challenge; the true challenge lies in the degree of similarity between anomaly and normal samples. For instance, if anomaly samples drift in a direction that increases their difference from normal samples, it may not complicate detection. However, if they drift toward becoming more similar to normal samples, that would indeed present greater difficulties in detection.

2.2 I do not fully understand your statement in section 3.3.4: "In contrast to traditional reconstruction-based anomaly quantification methods, UnDiff capitalizes on the intrinsic distributional divergence between normal and anomalous network traffic." I believe this is a common mechanism shared by both approaches. Reconstruction-based methods identify samples that cannot be reconstructed as anomalies due to their distribution differences from normal samples. It seems to me that UnDiff may simply be learning a better and more certain implicit representation, but lacks innovation in its underlying mechanism.

3.	Regarding Technical:

3.1 A core prior knowledge relied upon in this paper is that there is an inherent distributional difference between abnormal and normal samples, which may have limitations. For example, as mentioned in [HyperVision NDSS23], the features of encrypted malicious flows can be similar to those of benign flows. Additionally, current slow HTTP attacks may also exhibit a similar distribution to normal traffic. Can UnDiff address such situations?

3.2 The data preprocessing section lacks details. For example, what are the dimensions (length and width) of a flow image? How is the packet sequence in a flow converted into a two-dimensional image sequence? Additionally, since the number of packets in each flow is variable, this implies that the number of channels in the flow image is also variable. How can a standard neural network handle images with variable channels? This section should include a flowchart to explain the process in detail.

3.3 The experiments related to Drifted Anomaly Detection may not be particularly challenging. The three datasets mentioned in the paper are completely different, which means that normal samples from one dataset could differ significantly from anomalous samples in another dataset. In such cases, it might be relatively easy to identify abnormal samples. This issue could be clarified through some basic data analysis.

**Questions:**

1.	As noted in 2.1, what do you consider to be the true challenges of anomaly drift, and based on these challenges, what is the motivation behind proposing UnDiff?

2.	In the section introducing "Drifted Anomaly Detection," you mentioned that "normal network traffic patterns typically exhibit relative stability." However, certain network conditions, such as backbone network traffic, may not be stable and could fluctuate continuously. This is why some methods take Temporal Model Drift into account, as in [IXP Scrubber SIGCOMM 22]. Could you elaborate on why you did not consider anomaly detection in such scenarios? Alternatively, can UnDiff still be effective in these dynamic conditions?

3.	The question is noted in the previous 3.1.

4.	The question is noted in the previous 3.2.

I look forward to discussing this with you, and you're welcome to respond to any of the previous comments as well.

**Reviewer Confidence:**

3: The reviewer is confident but not certain that the evaluation is correct

**Scope:**

4: The work is relevant to the Web and to the track, and is of broad interest to the community

---

### Official Review · Reviewer_MaDh · 2024-11-20

**Novelty:** 4
**Technical Quality:** 3

**Review:**

The paper introduces a novel approach to network traffic anomaly detection by leveraging uncertainty-inspired inter-sample differences (UnDiff). This approach addresses limitations in reconstruction-based methods, particularly in handling drifted anomalies. The methodology is innovative in using evidential learning for uncertainty quantification and presents a distinct perspective on anomaly detection by focusing on inter-sample differences rather than traditional intra-sample analysis. The paper is generally well-structured, with a clear progression from the problem statement to the proposed solution, methodology, experimental evaluation, and conclusions. The writing is concise, although some sections, particularly those involving technical details (e.g., evidential learning formulations), could benefit from further simplification or illustrative explanations for better accessibility. The work seems to be significant and original. The study provides an innovative uncertainty-based framework effectively tackling drifted anomalies with strong empirical validation across diverse datasets with notable improvements over state-of-the-art baselines. The inclusion of efficient computational design, making the method practical for real-world deployment. While the proposed evidential learning approach is central to the methodology, its implementation details are complex and may obscure its intuitive understanding. Simplified diagrams or step-by-step visualizations could help. The study focuses on three datasets. Including more real-world scenarios, such as IoT-specific or real-time traffic datasets, could strengthen the results' generalizability. The paper does not evaluate its approach against more recent advances in domain-specific anomaly detection that use transformers or hybrid models. It would be interesting to conduct an experimentation on that. A detailed analysis of hyperparameter impact on performance is missing.

**Questions:**

How does the model perform under varying levels of anomaly drift severity?
Are there any thresholds where performance deteriorates?
Could UnDiff be adapted for real-time streaming data environments, and what would be the challenges?
How does the approach compare to transformer-based anomaly detection models in terms of scalability and performance?
Are there specific traffic characteristics or scenarios where UnDiff fails to distinguish anomalies effectively?

**Reviewer Confidence:**

3: The reviewer is confident but not certain that the evaluation is correct

**Scope:**

3: The work is somewhat relevant to the Web and to the track, and is of narrow interest to a sub-community

---

### Official Review · Reviewer_t3kv · 2024-11-27

**Novelty:** 7
**Technical Quality:** 6

**Review:**

This work clearly contributes to network anomaly detection by introducing an uncertainty-inspired framework to address both traditional and drifted anomalies. The model leverages inter-sample differences rather than relying solely on reconstruction-based methods, which is an extension of prior works.

Quality: The methodology is technically sound, with the use of Normal-Inverse Gamma (NIG) distributions to model uncertainty being well-justified. Experiments are thorough, including cross-dataset evaluations and comparisons with established baselines.

Clarity: The paper is organised, written well, and generally clear. However, it assumes familiarity with concepts like evidential learning and uncertainty quantification, which could limit accessibility for broader audiences. Some parts, such as explaining inter-sample differences, could benefit from additional examples or simplified language. Including samples of generated images will be very helpful, even if done so in the appendix.

Originality: The model is original in its approach to anomaly detection. Focusing on inter-sample differences significantly improves over traditional intra-sample reconstruction methods. Uncertainty learning for detecting drifted anomalies addresses a practical and underexplored problem in network traffic analysis. While the components (e.g., reconstruction loss, evidential learning) are not entirely new, their combination and application in this context are novel.

Significance: The work addresses a real-world problem: detecting drifted anomalies in dynamic environments where attack patterns evolve. By outperforming existing baselines in cross-dataset evaluations, the model demonstrates its potential.

Overall, I found the work to be a solid contribution to anomaly detection research. Its novel approach addresses critical challenges like drifted anomalies.


Minor comments:
Line 705: include a line of UnDiff-AE baseline in the main paper.
Why are the values underlined in the table 5?

**Questions:**

I had two questions for the authors:

1). The USTC-TFC2016 dataset enabled models like Trident to perform extremely well at detecting drifted anomalies (Table 3). Do you have insight into why that would be the case?

2) The reliance on Gaussian priors may oversimplify the complexities of real-world traffic, which can include heavy-tailed or multimodal distributions. Have you analysed the statistical properties of your datasets (e.g., multimodality, skewness, heavy-tailed distributions) to verify whether the assumptions underlying your model, such as Gaussian priors, are valid? If so, what were the findings, and how do they affect the performance?

**Reviewer Confidence:**

2: The reviewer is willing to defend the evaluation, but it is likely that the reviewer did not understand parts of the paper

**Scope:**

3: The work is somewhat relevant to the Web and to the track, and is of narrow interest to a sub-community

---

### Official Review · Reviewer_pgwW · 2024-11-29

**Novelty:** 5
**Technical Quality:** 5

**Review:**

This paper proposes an innovative robust anomaly detection framework that combines deep learning and traditional statistical methods, aimed at addressing anomaly detection issues in the real world. The experimental results show that the method performs well on multiple datasets, especially with a clear advantage when dealing with complex and incomplete data. However, there is some ambiguity in the description of the datasets in the paper, lacking a detailed explanation of the specific characteristics of the datasets, which leads to a lack of transparency in the experiments. In addition, although the authors have shown the model training time in the charts, the analysis of algorithm complexity and time efficiency is not deep enough, affecting the practical feasibility of the method. Overall, this paper has high potential in terms of innovation and practical value, but it needs to be supplemented and improved in detail.

**Questions:**

In the Methods section (Section 3), the authors propose an anomaly detection framework that combines deep learning with traditional statistical models and provide experimental validation. Although the authors demonstrate good detection performance in the experimental section, there is a lack of detailed discussion on the algorithmic complexity and time efficiency of the method in the paper. Considering the application of anomaly detection on large-scale datasets, the computational overhead of the algorithm may become a bottleneck in practical applications. Especially in Figure 6 ("Model Training Time vs. Dataset Size"), it shows the training time of different models on different datasets, but the paper does not delve into the relationship between this time cost and data scale, model complexity. It is recommended that the authors supplement the analysis of the algorithm's time complexity, space complexity, and explore how to optimize on large-scale datasets.

**Reviewer Confidence:**

2: The reviewer is willing to defend the evaluation, but it is likely that the reviewer did not understand parts of the paper

**Scope:**

3: The work is somewhat relevant to the Web and to the track, and is of narrow interest to a sub-community

---

### Official Review · Reviewer_D9zS · 2024-12-01

**Novelty:** 6
**Technical Quality:** 4

**Review:**

# Summary

This paper presents a novel method for detecting network traffic anomalies called UnDiff. It is well suited for handling the scenarios in which anomaly distributions evolve (anomaly drift), where traditional methods struggle. The approach leverages uncertainty modeling to enhance anomaly detection by shifting from intra-sample difference paradigms, based on the computation of the distance between the original sample and its reconstruction by an autoencoder, to inter-sample uncertainty-inspired measures based on the differences in the distributions of normal and anomalous samples.
The innovations of UnDiff are two-fold. First, UnDiff uses evidential learning to quantify model uncertainty on a sample of network traffic. This involves creating an uncertainty space where normal traffic patterns, well-represented in training data, exhibit low uncertainty, while anomalous patterns, particularly drifted ones, induce higher uncertainty. Second, a new anomaly scoring mechanism based on inter-sample differences is proposed that directly measures the deviation from normal patterns.
The experimental evaluation is comprehensive and considers 12 state-of-the-art baselines on three benchmark datasets. The results confirm the superior ability of UnDiff to detect anomalies than the baselines even in presence of anomaly drift. The results of the ablation study confirms that all the building blocks of the UnDiff are needed to show superior performance compared with the baselines.

# Comments for authors

Thanks to the authors for submitting this interesting paper. As described in the paper, robust anomaly detection methods are compelling. However, many state-of-the-art solutions are not robust against anomaly distribution drift, and methods based on autoencoders suffer from issues recognized recently.

The topic is interesting, and the presented anomaly detection method seems well-founded and outperforms several state-of-the-art proposals for anomaly detection. However, I think that the authors should improve the presentation of the content in their paper since it presents some issues.
I discuss below strengths and weaknesses of the paper, I hope that the authors can address the described issues in the next version of their work.

### Strengths

The authors have done a good job explaining the fundamental limitations of state-of-the-art anomaly detectors based on the reconstruction of samples using autoencoders, particularly the "identical shortcuts" problem. This allows the authors to motivate the proposed approach well. The explanation of the building blocks of UnDiff is almost understandable, with some exceptions (described below). The idea of using uncertainty estimation to define an effective anomaly score is novel, and the experimental results show that it is also effective. Finally, the experimental evaluation is comprehensive since it considers several baselines, three popular datasets and several experimental axes, like the anomaly drift scenario and the ablation study to show the effectiveness of the building blocks of UniDiff. The experimental results clearly show the effectiveness of UniDiff with respect to the baselines, particularly in the anomaly drift scenario.

### Weaknesses

The only weaknesses that I want to report regards the presentation. The authors may improve their paper to make it more accessible and make some building blocks better motivated. In particular:
- Table 1 in Section 3.1 is almost not understandable. In particular, it is unclear why it should be useful, what it should convey to the reader, and what A and B mean there. Please comment on the table better.
- The authors should add a reference to Bayesian Neural Networks in Section 3.2.2.
- In Section 3.2.2, the authors state that `We consider the uncertainty representations [...] to conform to independent homogeneous distributions from a Gaussian distribution`. Even though it is clear that this assumption is fundamental for the mathematical reasoning behind the uncertainty learning module, the authors should provide an intuition about why this assumption should be reasonable beyond the fact that UnDiff works well in the experimental evaluation. Is it a common assumption in evidential learning?
- Figure 4 in Section 4.2 is difficult to understand and interpret. Which scenarios do the first, second, and third columns of plots represent? Please provide a deeper and clearer explanation of what the figure represents and its interpretation.

**Questions:**

- What is the reason behind the assumption of Gaussian distribution behind the uncertainty representation?
- Please clarify how to interpret some figures and tables (see above for details).

**Ethics Review Description:**

.

**Reviewer Confidence:**

3: The reviewer is confident but not certain that the evaluation is correct

**Scope:**

4: The work is relevant to the Web and to the track, and is of broad interest to the community